# Female cyclists perceived effects and experiences of the menstrual cycle on training and performance

Louise Burnie[1]*, Paul Ansdell[1], Georgia Allen-Baker[1], Elisa Pastorio[1], Kirsty M. Hicks[1], Neil Heron[2], Natalie Brown[3,4]

1 Faculty of Health & Wellbeing, School of Sport, Exercise and Rehabilitation, Northumbria University, Newcastle upon Tyne, United Kingdom, 2 Centre for Public Health, Queens University Belfast, Belfast, United Kingdom, 3 Department of Sport and Exercise Sciences, Swansea University, Swansea, United Kingdom, 4 Welsh Institute of Performance Science, Swansea, United Kingdom

* louise.burnie@northumbria.ac.uk

## Abstract

The aim of this study was: (1) to understand female cyclist's experiences and perceptions of the menstrual cycle (MC) on training and competition performance, and (2) to explore female cyclist's comfort in having conversations relating to the MC with coaches and support staff. Twenty competitive female cyclists (age 35.1 ± 7.7 yrs, cycling for 11.9 ± 7.4 yrs, 3 elite, 7 sub-elite and 10 club cyclists) were interviewed using an open-ended, semi-structured approach. An inductive thematic analysis was conducted. Female cyclists' experiences and perceptions of the MC on training and competition performance were wide ranging with participants reporting a range of MC symptoms that varied in severity and timing in the MC. 45% of the cyclists had experienced irregular MCs either currently or in the past with participants suggesting possible causes may be due to low body weight and insufficient energy intake to support training. 73% of the participants spoke to their coach about their MC, but typically this was limited to a note on their training programme that they started their period, or a brief mention of how they are feeling. Participants highlighted the positive influence of elite athletes talking about their MCs in terms of accepting the influence of MC on performance and improving openness of conversation. Most participants said they lacked knowledge about the MC and how to manage MC related symptoms. The findings highlight the need for improved education on the MC in sport for cyclists and coaches to improve performance and athlete health and wellbeing.

## Introduction

The last five years have seen large growth in female professional cycling, with the number of Women's World Tour and UCI Pro teams increasing from 8 in 2020–22 in 2025 [1]. This includes the introduction of Paris-Roubaix Femmes in 2021, and the

**Data availability statement:** All relevant data are within the paper and its Supporting Information files.

**Funding:** The author(s) received no specific funding for this work.

**Competing interests:** The authors have declared that no competing interests exist.

reintroduction of an eight-stage Tour De France Femmes Avec Zwift to the professional women's calendar in 2022 after a 33-year hiatus. For the first time ever there was sex parity in all cycling events at the Paris 2024 Olympics [2]. There has also been an increase in the number of recreational female cyclists in the UK rising from 550,000 in 2013–1,023,271 in 2020 [3,4]. Despite the increase in professional and recreational participation, there remains a lack of evidence-based knowledge to effectively support female cyclists.

The changing concentrations of oestrogen and progesterone across the menstrual cycle (MC) exerts complex effects on multiple physiological systems [5,6] that could ultimately affect cycling performance [7]. A few studies have tried to quantitatively assess the effect of MC phase and changing hormone concentrations on cycling performance with conflicting findings, with some finding no effect of MC phase and others a significant difference between phases [8–15]. For females who take the oral contraceptive pill (OCP), several studies have found no difference in cycling performance for OCP users between inactive and active pill phases [8,15,16]. This approach of analysing females as a single cohort dismisses the influence of individual factors of the MC such as the occurrence and severity of MC/hormonal contraceptive (HC) symptoms, and their influence on perceived readiness to perform. For instance, evidence has suggested ovarian hormone concentrations across the MC do not appear to affect indoor time trial cycling race performance [17]. However, an increase in perceived negative MC symptoms on race day was correlated with an increase in indoor time trial time [17], therefore it is important to consider experiences beyond performance measures, including perceptual and individual changes.

Female athletes from a range of sports report suffering from MC related symptoms, including physical (heavy bleeding, tiredness, menstrual cramps and headaches) and psychological (changes in mood, increased anxiety/irritability) symptoms [18–24]. Athletes typically report symptoms to be worse around menses with symptoms reported to have a perceived negative impact on performance [19,21–23,25–29]. A greater severity of MC symptoms results in a greater likelihood of an athlete perceiving a reduction in their performance and increased time to recover, as well as increased use of pain medication and possibility of them missing or changing training or missing competition [18,23,30]. Athletes often choose to actively control their MC with HCs to reduce the perceived effect of menses on competition [19,30,31]. However, McNulty, Ansdell (23) found no difference in symptomology between naturally menstruating (eumenorrheic) and OCP users.

Female cyclists have not previously been included within research exploring experiences of menstrual symptoms. A recent survey of Australian female cyclists from recreational to elite level has found more than 80% of these cyclists reported that their MC-related symptoms impacted upon training and 41% made training adjustments based on these symptoms [32]. However, this did not explore how symptoms were experienced nor how they interacted with training and performance to inform future applied practice to support female cyclists. The variation in findings from recreational to elite is important to highlight and to continue to explore similarities or differences in experiences.

Beyond menstrual symptoms negatively impacting training and performance, MC disturbances are highly prevalent in female athletes, with up to 50% suffering luteal phase defects or anovulation, and a third may be amenorrhoeic [33] which has been reported to be even higher in endurance athletes (43%) [34]. MC disturbance is one of the symptoms of Relative Energy Deficiency in Sport (REDs) [35]. Female cyclists are particularly at risk of REDs due to high training volume and competition load [36,37], that they perceive low body weight is beneficial for performance [38], can have low energy intake insufficient for their training volume [39] and subsequently train with suboptimal energy availability [36]. Worryingly, it has been reported that only 22% of athletes in lean build sports would report amenorrhea [40]. Therefore, it is important to understand MC disturbance in female cyclists, and the factors they think influence this.

Several studies have found female athletes are often uncomfortable discussing their MC with male coaches, and may avoid talking about it altogether, even if their MC symptoms are impairing their ability to train and compete [19,25,28,32] or have irregular/absent menstruation [41]. Australian cyclists have said the coaching and management staff in cycling being predominantly male is a barrier to them openly discussing their MC [32], which could result in a lack of support for their wellbeing, limiting their long-term success [42,43]. The research on the comfort of athletes discussing their MC with their coaches has focused on elite senior athletes [19,25,28]. Consequently, there may be differences in sub-elite, amateur and junior athletes. Also, recently there has been more discussion of the MC in the media and by elite athletes [44], which may have influenced the openness of conversation between athletes and coaches compared previously reported.

To address the lack of knowledge of the lived experiences and perceived effect of the menstrual cycle on cycling training and performance, the aims of this study were:

(1) to understand female cyclists' experiences and perceptions of the MC on training and competition performance from an array of levels (club to elite) and ages (under 23 and senior).

(2) to explore female cyclists' comfort in having conversations relating to the MC with coaches and support staff.

## Methods

### Positionality

To address the research aims and to the understand subjective meanings and personal experiences of the participants in this study, an interpretivist approach was adopted. The research team recognise that reality is multiple and socially constructed and data (i.e., knowledge) is co-created between the researcher(s) and the participant. A combination of epistemological subjectivism and ontological relativism, to inform an interpretivism research paradigm was adopted [45]. Working within this paradigm, it is important to note that, the lead author is a female cyclist and considered an insider within cycling domains. This relationship with cycling communities helped with recruitment and to build rapport with the participants. The lead author/interviewer was viewed as a trusted individual as a fellow cyclist but also as a woman. Whilst this positioning had many benefits, it did create considerations from a reliability and trustworthiness perspective as her lived experiences of the MC and sport had the potential to influence data collection and interpretation – steps taken to minimise this influence are discussed in the data analysis section.

### Participants

Twenty competitive UK based female cyclists were recruited by purposive (criterion-based) sampling [46] based on (a) actively competing as a cyclist, (b) 18 years and older, (c) assigned biologically female at birth and (d) pre-perimenopause for both naturally menstruating and HC users. Participation level was categorised according to McKay, Stellingwerff [47] tier system, the participants consisted of three elite cyclists: tier 4 – competed for a Union Cycliste Internationale (UCI) registered cycling team (one of which rode for a UCI Women's World Team); seven sub-elite cyclists: tier 3 – competed in national level domestic competitions; and ten club cyclists: tier 2 – competed for an amateur cycling club. Participants

 

were age 35.1±7.7 yrs (range 21–45 yrs). Participants training history, characteristics and racing frequency are detailed in Table 1. The participants competed across a range of cycling disciplines including road racing, criteriums, time trials, hill climbs, indoor racing (e-sports), track, gravel, mountain bike and cyclocross. Eleven of the participants had a coach of which eight were male and three were female. We were unable to recruit any cyclists under the age of 21, despite contacting teams with junior and under 23 riders.

## Procedure

The study was approved by Northumbria University Faculty of Health and Life Sciences Research Ethics Committee (project number: 6056). Participants were recruited through the lead researcher's networks with National cycling federations and cycling teams and clubs. Participant recruitment was conducted from 23/02/2024 to 06/11/2024. Participants were provided with details of the study and written informed consent was obtained. The participants were sent a short pre-interview questionnaire (see supporting information: S1 File in S1 Text – pre-interview questionnaire) based on Brown, Knight (19), which took 10 minutes to complete to obtain demographic and MC information. Participants were also provided with a 18-item-symptom list to report their typical MC related symptoms [18]. We made two small changes to symptom names: changed stomach cramps to period cramps/pain & pelvic/uterine/ovarian pain, and mood changes/anxiety to mood changes/irritability/anxiety based on refinements made to the symptom list by Crawford, McNulty (48) to better describe the symptoms to participants. This symptom list has recently been used in MC related symptom research [23,48], and was used to provide information to assist the interview when talking to participants about their experiences of the MC. We did not exclude participants with any menstrual health conditions or those who have children.

The lead researcher conducted semi-structured interviews in English, with open-ended questions to allow participants to express their thoughts and expand on topics [45]. The interview guide was adapted from Brown, Knight (19) (see supporting information: S2 File in S1 Text – interview guide). Additional questions were added to the interview guide *'Do you experience any changes in your core strength, coordination, mobility, and flexibility throughout the menstrual cycle? Implications of this – training and competition, bicycle set-up'* which was informed by the findings of Brown, Knight (19). This could be an important factor for cyclist's as it could influence their ability to sit comfortably on a bicycle. The interview contained additional questions on bicycle set-up, injuries and saddle sores which is reported in [49]. The interviews lasted 26.1±7.0 mins (range 15.3 to 44.1 mins) and were conducted by the lead researcher either face to face or online via Microsoft Teams. Interviews were recorded using Microsoft Teams which also produced a transcript of the interview.

## Data analysis

The interview transcripts produced by Microsoft Teams were checked by the lead researcher for accuracy, removal of personal identifying information and where necessary, small grammatical changes were made to improve the flow of the text (see supporting information: S3 File in S1 Text – interview transcripts). They were then re-read by the lead researcher to ensure immersion in the data before undertaking analysis and coding of the transcripts using inductive reasoning in the

**Table 1. Participant training and racing characteristics (means,±SD) by Tier level (1-3).**

| Variables | Tier 2 -Club (10) | Tier 3 – Sub-elite (7) | Tier 4 – Elite (3) |
|---|---|---|---|
| Age (yrs) | 40±5 | 28±6 | 33±7 |
| Training background (yrs cycling) | 9±6 | 12±4 | 22±11 |
| Training frequency (number of sessions per week) | 7±3 | 7±1 | 8±1 |
| Training volume (hours per week) | 9±4 | 12±4 | 19±1 |
| Race volume (number of races per season) | 9±7 | 22±12 | 31±6 |
| Coach (number of participants (%)) | 3 (30%) | 5 (71%) | 3 (100%) |

software programme NVivo (QSR NVivo 14). A thematic analysis was conducted following the steps outlined in Sparkes and Smith (45) which is based on Braun and Clarke [50]. Phase 1: Immersion: the lead researcher (LB) immersed herself in the data by rereading the interview transcripts. Phase 2: generating the initial codes: the lead researcher systematically went through the interview transcripts and inductively coded the data into initial codes. Phase 3: Searching for and identifying themes: the lead researcher reviewed the initial codes then actively grouped them into main and subthemes which were decided by the researcher. Phase 4: Reviewing themes: the lead researcher then reviewed, reworked and refined the themes by repeatedly reviewing the generated themes, and the original data to check the themes generated captured the essence of the underlying data. Phase 5: Defining and naming themes: the lead researcher then reviewed the theme names to check they accurately and concisely summarised the essence of each theme. Phase 6: Writing the report: the lead researcher wrote the combined results and discussion sections using this as a further opportunity to refine the themes.

The research team acted as critical friends [51], by reviewing interview transcripts, initial codes and themes. This was not a linear process, and themes were refined throughout the process. Authors NB and LB discussed the themes resulting in refinement to groupings and theme names before the final themes were agreed. The whole research team reviewed and confirmed the final themes presented within this paper.

As discussed earlier, and in the interest of reflexivity, the lead author/reviewer was viewed as an insider within female cycling domains and this brought many benefits but required careful reflection throughout the study (i.e., during conception, interview schedule design, data collection and data analysis phases). Critical friends within the research team (four females and two males) were utilised throughout the research process to encourage the lead researcher to think reflexively about her analysis. We acknowledge that the lead researcher acting as the primary data coder and analyst may limit the interpretive breath. However, we tried to minimise the influence by using the research team as critical friends. Data interpretation was tracked throughout the study by the lead author. This functioned as an important audit trail providing transparency throughout.

## Results and discussion

Self-reported descriptive MC data and HC use are displayed in Table 2; and MC related symptoms for the naturally menstruating and non-hormonal (intrauterine device – copper coil) participants are reported in Table 3. Through thematic analysis five main themes (MC characteristics, disturbance and HC use; MC symptom interaction with training and competition; Coping and management of MC symptoms; MC conversations; Knowledge and education on MC) and 22 subthemes were developed (Table 4). Each of these are discussed below.

### MC characteristics, disturbances and HC use

Twelve participants were naturally menstruating (60%), six participants were currently using HCs (30%), and two participants had an intrauterine device (10%) (Table 2). The reasons for participants using HCs were for birth control, to manage MC symptoms, eliminate/control bleeding or to treat acne:

*'I have always struggled with quite extreme cramps, so I went on the combined pill very early probably when I was 13, more to combat those symptoms than anything to do with contraception.' (P15 - club)*

For three participants, the decision to use HCs was to manage their MC around competitions to reduce the negative effect of the MC, sometimes on the recommendation of coaches or doctors. This is consistent with athletes across a range of sports [19,25,30–32]:

*'When I was on a cycling talent team [as a teenager], they [my coaches] recommended I went on the pill, so that I could control it [my MC] for national championships, which looking back that's not good.' (P18 - elite)*

**Table 2. Self-reported menstrual cycle status characteristics.**

| Menstrual cycle history | | |
|---|---|---|
| Mean age of menarche (years) | | 13±2 |
| Naturally menstruating (number (%)) | | 12 (60) |
| Hormonal contraceptive users (number (%)) | | 6 (30) |
| Non-hormonal contraceptive (intrauterine device) (number (%)) | | 2 (10) |
| **Naturally menstruating group** | | |
| Menstrual cycle length | Short (≤ 20 days) | 0 |
| | Regular (21–35 days) | 11 |
| | Long (≥ 36 days) | 0 |
| | Amenorrhea | 1 |
| Average duration of menses | 2-5 days | 8 |
| | 5-7 days | 2 |
| | 7+days | 2 |
| Athletes tracking their menstrual cycle | Smartphone app | 7 |
| | Manually | 3 |
| | No | 2 |
| Previous hormonal contraceptive use | Combined oral contraceptive pill | 4 |
| | Progestin-only oral contraceptive pill | 5 |
| | Contraceptive injection | 1 |
| Reason for stopping hormonal contraception | Adverse side effects | 4 |
| | No longer wanting to control hormones | 3 |
| | No longer needed for birth control | 2 |
| | Medical condition | 1 |
| **Hormonal contraceptive group** | | |
| Type of hormonal contraception | Combined oral contraceptive pill | 3 |
| | Progestin-only oral contraceptive pill | 1 |
| | IUS (intrauterine system) | 1 |
| | Other (Hormone Replacement Therapy) | 1 |
| Duration on hormonal contraception (years) | | 9±12 |
| Experience withdrawal bleeds (number) | | 6 |
| Previous use of different hormonal contraception (number) | | 4 |
| Reason for hormonal contraception use | Birth control | 3 |
| | Manage menstrual cycle symptoms | 3 |
| | Eliminate/control bleeding | 1 |
| | Other | 3 |
| **Non-hormonal contraceptive (intrauterine device) group** | | |
| Menstrual cycle length | Short (≤ 20 days) | 0 |
| | Regular (21–35 days) | 1 |
| | Long (≥ 36 days) | 1 |
| | Amenorrhea | 0 |
| Average duration of menses | 2-5 days | 2 |
| | 5-7 days | 0 |
| | 7+days | 0 |

*(Continued)*

**Table 2.** (Continued)

| Menstrual cycle history | | |
|---|---|---|
| Athletes tracking their menstrual cycle | Smartphone app | 2 |
| | Manually | 0 |
| | No | 0 |
| Previous hormonal contraceptive use | Progestin-only oral contraceptive pill | 1 |

**Table 3. Self-reported symptoms in the naturally menstruating and non-hormonal contraceptive groups.**

| Symptom | Number of participants | % of participants |
|---|---|---|
| Changes to/difficulties breathing | 2 | 14 |
| Nausea, sickness & vomiting | 3 | 21 |
| Constipation | 3 | 21 |
| Dizziness/light headiness/reduced co-ordination | 5 | 36 |
| Poor concentration/memory | 4 | 39 |
| Joint pain/muscle aches & cramps | 6 | 43 |
| Temperature fluctuations | 9 | 64 |
| Disturbed sleep | 5 | 36 |
| Diarrhoea | 6 | 43 |
| Headaches/migraines | 1 | 7 |
| Lower back pain | 8 | 57 |
| Water retention | 5 | 38 |
| Bloating/increased gas | 13 | 93 |
| Period cramps/pain & pelvic/uterine/ovarian pain | 13 | 93 |
| Tiredness/fatigue | 10 | 71 |
| Breast pain/tenderness | 7 | 50 |
| Cravings/changes in appetite | 9 | 64 |
| Mood changes/irritability/anxiety | 10 | 71 |

Some of the OCP users reported using an extended OCP regimen (no break between active pills) to manage when their withdrawal bleed occurred around competition and training, this behaviour has also been reported in Australian cyclists [32]:

*'The doctor said, it's fine to do that [use combined pill packs back-to-back] for sports and managing life where it's convenient I'll take a break.' (P19 - club)*

Eleven of the participants had previously used HCs. One of the reasons for stopping HC use was the adverse side effects of HCs (four participants) especially related to the progestin-only OCP, which included breast pain, spotting and mood changes which have previously been reported as side effects [52]:

*'I had side effects for the two progesterone types. I had constant boob pain. I was like, I can't do that anymore. And then the other one was spotting.' (P8 – sub-elite)*

*'I was not myself [on the progestin-only pill]. I went really negative, and I couldn't control my emotions very well.' (P14 – elite)*

**Table 4. Themes.**

| Main themes | Sub-themes |
|---|---|
| MC characteristics, disturbances and HC use | HC use |
| | Irregular menstruation |
| MC symptom interaction with training and competition | Heavy bleeding |
| | Symptoms |
| | Training considerations |
| | Competition perspectives |
| | Awareness |
| Coping and management of MC symptoms | Medical |
| | Menstrual products |
| | Mentality |
| | Planning |
| | Diet changes |
| MC conversations | Coaches |
| | Teammates |
| | Factors influencing conversations |
| | Prevalence of conversation |
| | Role models |
| Knowledge and education on MC | School |
| | Self-educated |
| | Coach/Team/Sports Science Practitioners |
| | Education provision |
| | Future research and knowledge |

Another reason for stopping HC use was participants no longer wanting to control their hormones and have a natural MC (three participants). Also, some participants no longer required HC for birth control (two participants) or stopped on medical advice (one participant). Often when participants stopped using HC it took a long time for regular MCs to return, or resulted in increased symptoms initially during menses:

*'I was on the pill for quite a long time and then when I came off it, I didn't have any periods for a really long time which I had explored medically.' (P20 – club)*

*'When I came off the pill for… three or four months I wouldn't ride my bike that day [first day of period] because I was in bed. I was in quite a lot of pain.' (P4 – sub-elite)*

The adverse effects of HCs, along with the different HC options need to be communicated with athletes, so they can make an informed choices on whether to use HC. This is supported by recent research which highlighted females would like to receive more information from healthcare professionals regarding the risks of HCs and the alternatives [53]. However, healthcare professionals have reported there are several barriers to achieving this which are: time constraints, lack of resources and lack of standardisation in information and guidelines for prescribing HCs which need to be addressed by the healthcare systems [53].

The HC use (30%) in this study is lower than reported in Australian cyclists (44%) [32]. Although this study has a small sample size, the HC users were spread across all three performance levels which is in agreement with previous research that HC use is similar across performance levels from amateur to elite athletes [32,54]. It has been suggested that the lower rates of HC use in cyclists is because it is an individual sport and they can tailor training to their individual needs

 

[32]. Another reason suggested by some participants was they wanted to be more in tune with their body by having a natural MC (they were previous HC users), which agrees with anecdotal evidence of an increasing number of Australian cyclists being naturally menstruating [32]. Lee, Peiffer (32) suggested this is due to elite athletes now regarding a regular MC as a sign of good health. Our participants may also be following the wider societal changes which in recent years, has seen reduced HC use in the general population [55,56].

Nine participants (45%; two elite, three sub-elite and four club cyclists) reported that they either currently had irregular menstruation or had a duration of irregular menstruation in the past. This is in agreement with previous research that levels of MC disturbance do not differ across competition levels [30]. Our study has similar percentages of participants suffering MC disturbance as endurance runners [34]. Endurance sport athletes are considered greater risk of MC disturbance due to high training volume and the focus on lean body mass for performance [36,57]. Seven of these participants suggested their irregular menstruation were potentially due to low body weight and hard training (participants trained 11.6 ± 4.8 hrs per week, with a maximum training volume of 20 hrs per week), with insufficient energy intake. Both of these are risk factors for developing MC disturbance and REDs [58]. This agrees with previous research that female cyclists are body weight conscious and see low body weight as beneficial for performance [38].

*'When I got into cycling, it's those couple of kilos lighter that makes a difference. When I would have the off season and I would take some time off my bike, that's when I'd get a period or if I was injured and had some weeks off then I'd have one. So, I think it was a case of lack of energy availability really.' (P10 - elite)*

*'I lost weight, started cycling a lot more, training a lot more, and then I just didn't have a period for many years but what I remember when I did have them they were always very light. I've never really had periods.' (P14 - elite)*

*'I did go a year without having a period and I think that was due to training hard and not eating enough to support the training but thankfully I managed to get it back and since then my period has been regular.' (P11 – sub-elite)*

One elite rider commented that she thought most of her teammates probably did not get regular menstruation:

*'I think that there's probably a lot of the team that don't have regular periods actually.' (P10 - elite)*

Concerningly, several participants reported that some of their coaches and medical practitioners see irregular menstruation as normal for athletes which normalises irregular menstruation and prevents athletes seeking support or treatment [30,41].

*'It was my first year with the coach and he didn't seem to think that it was a problem. He just thought, OK, she's getting lighter, she's getting faster. So, not having a period, it's fine really and it's normal for female athletes, which it definitely isn't.' (P11- sub-elite)*

*'Maybe after a year or a year and a half I went to the doctors, and they did a check up and said it's normal [irregular periods]. They say you're an athlete, the only way to get them [periods] back is to stop cycling because over my time of being an athlete and not having a period I've put weight on, and I haven't had them back, so I think it's because you just push your body so much.' (P14 - elite)*

Some participants mentioned negative side effects of having irregular menstruation which included increased injuries, reduced bone density and fertility problems, which are all symptoms of REDs [35,59]:

*'My DEXA scans showed slightly dropping numbers, a downward trend in bone density, which obviously cycling doesn't help, but also a lack of periods or energy availability doesn't help either.' (P10 - elite)*

Also, one participant reported a sports gynaecologist suggested going on the OCP to treat lack of menstruation:

*'He said, look, I think if you're not going to get periods when you're pushing your body like this, being an athlete, then you should probably try and get them with the pill, because then you'll have all the hormones…. He said it's a good thing for athletes to do that.' (P14 - elite)*

### MC symptom interaction with training and competition

Five participants (one sub-elite and four club cyclists) reported heavy bleeding during menses, only one participant said it affected their daily life and training and therefore used HCs to manage. Heavy bleeding has been reported to impact performance and increase the risk of anaemia [24].

There was a range of MC symptoms reported (Table 3) and further discussed during the interviews that varied in severity and timing which agrees with the findings of previous research [19,25,60]. The variability highlights the individualised nature and therefore approach to symptom management for female athletes [61].

Across the participants in the naturally menstruating and non-HC groups 93% reported experiencing period cramps and pelvic/uterine/ovarian pain (Table 3) which is similar to previous research [23], and has been the most commonly reported symptom in cyclists, with three quarters suggesting period pain affects training [32]. Typically, period pains and cramps were worse on the day before and first few days of menses (reported by 80% of participants). 93% of participants also reported increased bloating and gas, with 71% reporting tiredness and fatigue, mood changes, irritability and anxiety (Table 3). The symptoms reported by the participants varied:

*'During my period, I think I am quite lucky in terms of symptoms. I have abdominal pain at the start and then not much else, really.' (P9 - club)*

*'I've had a couple of races that have fallen on day one [of my period]. I feel like I'm asthmatic on that day, literally I can't breathe and that's the most annoying thing in terms of cycling performance impact.' (P6 – sub-elite)*

Nine participants reported symptoms before their period including period pains, tiredness, food cravings, mood changes (more emotional and irritable), dizziness and disrupted sleep. For example,

*'About five to seven days before I start [my period] I have PMS [premenstrual syndrome] symptoms. I lose my appetite a bit and the symptoms can be that I'm feeling dizzy, a bit lightheaded, sluggish, and struggle to sleep.' (P14 – sub elite)*

*'I find that the worst symptoms are premenstrual. So, when I'm actually on my period itself I don't feel too bad. It's maybe the 10 days to week leading up to it, really bad cramps and lots of low mood and things.' (P3 - club)*

*'Normally I can control what I eat, but I'm like just give me sugar, no control the day before [my period starts]. I just want chocolate.' (P2 - club)*

Three participants reported negative symptoms (cramps, feeling more emotional, reduced power output when cycling) around ovulation:

*'Bizarrely, it's usually about 24 hours. I know myself that if in my training if I've got efforts that day I either say to my coach can I just have endurance or I just suck it up and know that I'll lose 30 watts… I'm like the world's falling apart because I feel really emotional. I plan around it.' (P18 - elite)*

Nine participants reported their MC, and symptoms had changed with age, in some cases due to them taking and then stopping HCs, and also being more self-aware of their MC as they got older which has previously been reported [19]:

*'I think maybe I'm just more self-aware… I'd say in my 20s, I didn't really notice, didn't really affect me at all.' (P6 – sub-elite)*

Half of the participants tracked their MC (six club, three sub-elite, one elite cyclists) which is similar to Australian cyclists where 58% reported tracking their MC [32]. However, participants in this study only tracked when their period started and put basic comments on how they were feeling on their training programme; this did not differ between performance levels. None of the participants reported tracking MC symptoms even though for some participants these symptoms affected training and competition.

Thirteen of the participants (two club, five sub-elite, two elite cyclists) modified their training around the MC, including to avoid the need for toilet stops to change menstrual products on outdoor rides during menses by doing shorter training rides outside or training indoors.

*'It [bleeding] does definitely become a bit of an itinerary for the day. I've got to think about leaving it as late as possible before going out and then timing it. My long rides aren't really more than 4.5 hours, so I literally [change menstrual product] before I go out and then if I get back within the 4.5 hours, I tend to be OK.' (P1 – sub-elite)*

*'If it was as heavy as it can sometimes get, then there might be a day that I would say I couldn't go because [no toilets] just side of the road. When you are with guys you feel a bit more conscious… it'll take me ages to get my kit off and then on, so I just don't bother normally.' (P13 – club)*

*'I never want to do a group ride or anything in the first couple of days [of my period], so when we're [cycle] touring that's always a problem as well when you're out all day and you have to factor in where there might be a toilet, so it is a deterrent.' (P9 - club)*

Another reason for reducing long outdoor rides was increased saddle discomfort during menses, as it has been reported labial sensitivity increases during bleeding (Lee et al., 2024).

*'I do find that longer rides are more uncomfortable when I'm on my period. I find that in general I'm a little bit more tender and the saddle is a bit more uncomfortable when I'm on my period. So, I don't tend to do longer rides at that time.' (P15 - club)*

Training indoors can allow participants to easily adapt or stop a session if they had negative MC related symptoms:

*'When I was training a lot more outside, I'd prefer to stay indoors if I had period symptoms, because then I knew that I'd be able to stop if I didn't feel well, whereas it's more difficult to do that out on the road.' (P11 – sub-elite)*

Five participants (one club and four sub-elite cyclists) reported occasionally missing training due to MC symptoms particularly if they suffered bad cramps or heavy bleeding which supports the findings of Lee, Peiffer (32) who reported a quarter of Australian cyclists skipped training due to period pain.

*'I get quite bad cramps. It's intermittent - some cycles it can be absolutely crippling and other ones not so bad. I can't get on with things. Couple of painkillers, need like half an hour for a painkiller to kick in and I'm just curled up, literally walking around is just too sore. I couldn't even imagine riding around on a bike, so that would be the only time in which it would really get in the way.' (P1- sub-elite)*

Most participants continued with their training as programmed and accepted it would feel harder or they would reduce their power targets, with a feeling they did not want to miss out on training. This mindset has been reported in other sports [19,25].

*'I never used to moderate it [training] because otherwise it felt like it was cutting out quite a lot of your training time.' (P9 - club)*

Two participants felt their MC has no effect on training or performance as they either have very irregular menstruation or have no symptoms from their HC use.

There are mixed messages on social media about training to MC phases and some participants would like to know if they should adapt their training to their MC to improve performance:

*'I refuse to change my training. I really dislike these people on social media that say you need to change your training in and around your menstrual cycle. I don't agree with that. I feel that if it makes you feel good, do it.' (P18 - elite)*

*'It would be interesting to understand more in terms of how much I should or could improve my training by adapting to where I am in my menstrual cycle, because at the moment it just sort of happens and I crack on with the training that I had planned and try and make the best of it.' (P15 - club)*

Only one participant reported adapting training based upon predicted phases of the MC:

*'I have done research around the menstrual cycle, so I do train with it a bit and my coach is pretty good with that. So, we have a recovery-ish week in the build-up [to my period].' (P8 – sub-elite)*

Several participants felt coaches offered generic training – coaching females as "small men" '*I find many coaches; they just say that women should train like men which definitely isn't the case.' (P11 – sub-elite)*' – and that their training programmes did not account for the known sex differences in physiology and response to training [62]. This highlights the need for improved coach education on coaching female athletes, whilst also acknowledging the need to support coaches in understanding the biopsychosocial interaction of the MC and cycling training/performance.

*'I had a male coach a few years ago. I remember speaking to him about that [training around MC phase]. I wanted to introduce it as a bit of a trial run and I was his only female athlete, and I found a lot of the training wasn't tailored, it was very generic. So, I don't feel like coaches really take it into context and use it to benefit the athlete. I did want to do it with him, and it just never really took off. I've had female coaches since then and it's just never been brought up.' (P1 – sub-elite)*

Currently there is insufficient high-quality evidence to support MC phase-based training [5,61]. Indeed, the role of sex hormones on factors such as metabolic thresholds [10], substrate metabolism [63], and oxidative function [8] during exercise is thought to be minimal. For this reason, individualised training informed by athlete symptoms and perceived effects of the MC is recommended [61,64]. Importantly, the absence of consistent group-level effects does not preclude meaningful inter-individual variability. Indeed, while hormonal fluctuations may not systematically influence performance outcomes across populations, some athletes may still experience noticeable changes linked to their MC, which is practically relevant in applied sport settings. This is supported by the findings from this study as female cyclists had varied symptoms and individualised experiences their MC in relation to training and performance. Lee, Peiffer (32) recommended female cyclists could benefit from phase controlled training based on MC symptoms and perceived negative effects on performance and two thirds of cyclists they surveyed believed training should be phase controlled [32]. However, the performance literature provides inconsistent evidence for changes in strength [65–67] and aerobic capacity [8,10,63] across the MC and therefore may not benefit cycling performance. In addition, barriers exist to the implementation of this approach including athletes' lack of comfort in having conversations with their coaches about their MC and athletes not systematically tracking

their symptoms across the MC, alongside biological measures to confirm phases. To start, athletes and coaches could track MC symptoms over several MCs to understand individual symptom profiles to inform strategies to mitigate training and performance variations across the MC.

In terms of off the bike training, several participants reported their balance was worse in the pre-menstrual phase and during menses which other elite athletes have reported [19], and is supported by findings of empirical studies where dynamic and static balance was poorest during menses [68,69].

*'Balance is completely off around being on [my period].' (P6 – sub-elite)*

80% of participants (including naturally menstruating, HC and intrauterine device users) reported negative effects of their MC or HC on their performance. Participants perceived sessions to be harder, have reduced motivation and for some participants reduced power output and increased heart rate, particularly during the first few days before and the start of menses. However, this was not universal as some participants perceived they competed well during menses. This agrees with previous findings that there is a varied perceived effect of MC phase on performance which is consistent across performance levels from recreationally active to world-class [19,70]. Cyclists' physiological tolerance of workouts and perception of effort in particular high intensity interval sessions have been reported to change with MC phase and symptoms [15,32]. It has been reported in cyclists that an increase in perceived negative MC symptoms on race day correlate with an increase cycling indoor time trial time [17].

*'One 10 mile TT [time trial], I felt awful. I didn't want to go. I was in a terrible mood and then it felt really, horribly hard. A lot harder than a normal 10. Then I went to the loo to get my number off my suit, and I was like, oh, [my period had started]' (P17 - club)*

*'I think I was quite lucky when I won a National Championship, that was the day before my period started, so that was really the worst possible time and actually when I trained on the hill the weekend before I put out higher power than I did on the day of the race.' (P9 - club)*

Participants reported certain MC phases can have a positive effect on performance, but the phase varied between participants:

*'After [my period] maybe for like the next 10 days I am quite happy, quite motivated, feel more motivated than I would other times.' (P6 – sub-elite)*

*'I do tend to notice that I am much stronger [during my period], so I tend to do things like weightlifting.' (P15 - club)*

Some participants were unsure or hadn't analysed the exact effect of their MC on performance.

*'I've just never really analysed the impact of that [MC] to be honest.' (P19 - club)*

Whereas, for other participants their awareness of the impact of their MC has increased:

*'Knowing how I started to feel in races … I'm a bit more mindful of it [my MC] now.' (P2 - club)*

In agreement with our findings, varying levels of self-awareness on the perceived effect of MC on performance have been reported among athletes [71]. For athletes experiencing symptoms and potential negative interactions with training and/or performance, tracking of the MC, along with education, may help improve the management of the MC to optimise health and performance.

   

## Coping and management of MC symptoms

Pain relief medication is by far the most common strategy (13 participants) to manage period pain which agrees with previous research [19,21,25,30]. Participants did not use non-pharmacological strategies to manage dysmenorrhea even though there is some evidence to suggest exercise, acupuncture [72] and nutrition [73] could be effective.

*'I'd say [I take painkillers] probably nearly every four hours on day one. And then on the second day probably a couple of times, as the day goes on it would start to then ease.' (P1 – sub-elite)*

*'I still usually have probably a day, day and a half during my cycle where paracetamol or ibuprofen is absolutely my friend.' (P15 - club)*

Three participants used HC to manage their MC around competition and to control heavy bleeding:

*'It was more to start with to manage my period, so if I had a competition … that I didn't want effecting, I could control when I had it.' (P16 – sub-elite)*

45% of participants in this study reported either they had irregular menstruation or had irregular menstruation in the past. One sub-elite cyclist had to get private medical treatment to help her return to regular MCs which involved working with a nutritionist to fuel for training properly:

*'I think under fuelling and training, maybe after two years they [periods] came back, I got some support to get them back, the NHS were pretty useless. So, I went to see somebody else privately, and then a couple of years ago they did the same, which they [periods] must have gone for about six months. I think when I get stressed, I just tend to under fuel, but I'm working with a nutritionist now and I'm just so much more aware of what you need to get into your body to do what we do as cyclists.' (P6 – sub-elite)*

One participant mentioned they had discussed their irregular menstruation with their coach with the aim to try and return to regular MCs to improve performance.

*'A goal was to try and have a regular period because it's good for performance. You will perform better as an athlete.' (P10 - elite)*

In agreement with findings from this study, it has been reported that athletes of all competition levels can find accessing specialist support for MC disturbance, low energy availability and REDs difficult or they receive incorrect advice [41]. The International Olympic Committee published a new consensus statement on REDs in 2023 [59]. In this statement they discussed the need for prevention of REDs by improved education of athletes, coaches, sports teams, sports science practitioners and doctors on prevention of REDs, the symptoms and long-term health consequences. However, based on the responses by participants in this study, which highlighted their lack of MC education as well as lack of knowledge within coaches and medical staff, further work is required to improve support for athletes at risk of MC disturbance and REDs.

Several participants reported using menstrual cups which can be very useful as they do not need to be changed as regularly as other menstrual products and have been reported to be used by females in other outdoor sports such as climbing which lack toilet facilities [19]:

*'I use a menstrual cup which is an absolute game changer.' (P18 - elite)*

*'I use menstrual cups. I had to use a tampon a couple of months ago because I forgot my menstrual cup wherever I was, and I was like this is horrible.' (P8 – sub-elite)*

Participants had different psychological approaches to competitions when they had negative MC related symptoms. These psychological approaches were acceptance (mentioned by three club, three sub-elite and one elite cyclists), adjusting of race goals, or just getting on with it (mentioned by five club and five sub-elite cyclists). This participant highlights the acceptance of negative MC symptoms and potential impact on performance, consequently the adjusting their race goals:

*'I think if it was a big race like the indoor cycling world's and I had PMS symptoms, I wouldn't miss it, but I would adjust my expectations. So, I'd go into the race and instead of aiming for a top 20, I'll just aim for maybe the top 40 or I could help a teammate instead, try to adjust my goals based on how I feel.' (P11 – sub-elite)*

Athlete acceptance of the possible impact of their MC has been highlighted previously as a coping strategy [19,71]. In cycling where cyclists often race in teams, their role in the team could change if they are experiencing negative MC related symptoms, such as riding in support of a teammate.

Many participants across all performance levels had the mentality of "just get on with it" when it came to training and competition which has been highlighted as one of the coping strategies used by athletes [71].

*'I just get on with it. I think it is just that mentality, especially if you've paid for it [race entry].' (P4 – sub-elite)*

In some cases, participants mentioned they didn't know what else they could do to manage their symptoms.

*'Because it's really difficult when you're competing and if a race is going to fall at the wrong time. You can't ask a national event to move because of your cycle. So, I don't know really. I don't know what options there are or what there is that can sort of alleviate symptoms.' (P9 - club)*

A few participants planned racing around their MC (particularly if they race regularly indoors).

*'So normally I wouldn't race in that week [week before my period]. So, I try to plan my racing in a way that I don't need to race in that week. I didn't used to do that. It's only the past year I've started doing it because when I race, I like to feel like I'm 100% in good shape and if I know I can't perform at my best I prefer to not race.' (P11 – sub-elite)*

They may also adapt their training programme, to avoid long rides during menses and adjust their position on the bike due to discomfort, these factors are specific to cycling:

*'It does feel a little bit uncomfortable [sitting in TT position], but it's not something I've had to do a lot because if I'm training on that day, I just sit up, I won't worry about it too much. It's been a very rare occasion that I've had to actually race in that position.' (P20 – club)*

Several participants who suffered disrupted sleep planned to go to bed early and accepted they may not sleep as well but they are recovering:

*'Just going to bed earlier … so, I've got more chance of sleeping for longer.' (P6 – sub-elite)*

Three participants modified their diet around their MC, typically increasing energy intake during menses, which has been reported previously [74]. Only one participant (sub-elite cyclist) mentioned taking any supplements (magnesium) to help with premenstrual symptoms. A systematic review by Brown, Martin (73) found there is some scientific evidence to support taking magnesium may be beneficial for reducing dysmenorrhoea when taken in conjunction with calcium but research evidence is of low quality and inconsistent.

In general there was a lack of awareness by participants in this study of the non-pharmacological treatments for MC-related symptoms which can include cognitive behavioural therapy relaxation techniques, exercise and supplements [75] even though 75% of the naturally menstruating females reported premenstrual symptoms. There was a lack of strategies considering the psychosocial impacts reported of the MC on training and performance, potentially related to lack of education and awareness.

Tracking of menstruation, MC length and MC related symptoms has been recommended by previous research to investigate if and/or how females are affected and to plan and adapt accordingly if necessary [61,64]. Rugby players have reported the personal benefits of tracking their MC in improving their knowledge and understanding and allowing them to plan strategies to better manage their MC [76], this required further investigation if the same benefits are seen with female cyclists in an individual sport compared to team sport environment. If tracking is implemented, ethical considerations on how MC tracking information is used by coaches and practitioners is very important to athletes and it needs to be agreed how the data will be used in advance of implementing MC tracking [48,77], as female team sport athletes have expressed concern that this data could be used to inform team selection [78,79].

## MC conversations

Despite ten of the participants reporting, they were comfortable talking to their coach or personal trainer about their MC, often this information was shared indirectly via comments in Training Peaks (training planning and recording software). Of the cyclists who had a coach, two of the three elite cyclists, four of the five sub-elite cyclists and two of the three club cyclists spoke to their coach about their MC. However, for most participants, the comments were limited to a note on their training programme that they started their period, or a brief mention of how they are feeling:

*'I always put it in Training Peaks... leading up to it [my period] or the actual day … so that he can see when it's happening as well.' (P20 -club)*

Three participants did not discuss their MC with their coach, even though the participants had suffered either MC disturbances, MC related symptoms that they perceived to affect their training or HC use side effects. Six of the participants (club and sub-elite cyclists) which includes those who had coaches in the past but are not currently coached, said they hadn't or don't talk to their coach about their MC: either because their coach hasn't asked, or they were uncomfortable in having this discussion as their coaches were male and the MC is perceived as a taboo topic that's not really talked about across all levels of cyclists, and in some cases would just say put a generic comment that they were feeling rubbish and not specifically say this was related to their MC:

*'It definitely was not talked about within cycling with coaches.' (P18 - elite)*

*'Our team manager is male as well and I don't think I would [mention if I had period symptoms] … I think it's a generational thing… I think they feel it's a bit of a taboo or it's an embarrassing thing to talk about.' (P1 – sub-elite)*

*'So, I have been coached in the past at school and at university, but it's never ever been raised or talked about. My coach has always been male. I don't know if that's had [an effect]. People find it awkward to talk about. A coach has never spoken to me about it.' (P12 - club)*

Aligning with findings from other sports such as field hockey [48], two factors influenced comfort of conversation with their coach about their MC: a) the closeness of the coaching relationship; and b) the gender of their coach.

*'I think because I'm so comfortable with my coach now, I wouldn't second guess ever mentioning anything. But I think if it was someone new or maybe a male, I'm not sure if I would bring it up initially.' (P16 – sub-elite)*

Cycling is a male dominated sport as most coaches, team managers and support staff are male [32]. In this study, of those cyclists with a coach, 73% of the coaches were male, which is representative of the qualified cycling coach population in the UK [80]. Female athletes have reported having male coaches and support staff can make them feel less comfortable discussing their MC [19,25,28,32], which was also reported by participants in this study. Male coaches in particular can be embarrassed or feel it is a taboo topic in society [81], which reinforces the difficulties females feel in talking about their MC with coaches. This highlights the need for coach education to provide guidance to coaches of all genders on how to approach this topic with their riders and create a supportive environment [82–84]. This was highlighted by participants who talked about the strength of the coach-athlete relationship as a factor that positively affected them having discussions about their MC. It has also been suggested that female facilitators could help [83], however this is challenging in cycling as it is a male dominated sport.

In contrast most participants were comfortable talking to their female coaches and their teammates about their MC symptoms, particularly if they were part of an all-female indoor racing team:

*'When I was younger, I never talked about the menstrual cycle. Maybe also because they were males. But I had for two years a female coach and with her I never had problems with the menstrual cycle, but some of my teammates had and so they felt free to talk with her - it was easier to talk with a female coach about their period.' (P7 – sub-elite)*

*'My teammates are really good. So, we speak to each other quite often actually. If I have to miss a race, I'll just say to them, I'm sorry I've got bad period symptoms and then they're very understanding because you know quite a few of them are in the same boat as me.' (P11 – sub-elite)*

Participants felt there was more discussion and awareness around the MC now compared to in the past when they were junior athletes and that this is a positive step. This is part of a wider societal change with increased media coverage around the MC and the feeling that menstrual stigma has decreased [44]. Younger females have increased awareness around health and social issues and feeling more comfortable sharing their experiences [44]. This may also account for the greater number of participants in this study (73%) who discussed their MC with their coach which is higher than previous findings from Australian cyclists where 51% discussed their MC with their coach [32] and in Australian athletes where only 25% spoke to their coach [28].

*'I feel certainly in the last 10-15 years it's been much more topical than when I was 15. There's so much more understanding now or encouragement actually to track it, to understand when you feel good and why you feel good at different points in the cycle and understand what your hormones are doing and when it's a good time to train in a certain way or why you might not feel good. I think there's been a huge increase in the understanding and the encouragement for athletes to try and not necessarily use it to our advantage, but to prevent it from being a disadvantage.' (P10 - elite)*

*'I think it is better that people are more open about these things now because historically when I first started competing, nobody spoke about it [MC or periods] at all.' (P9 - club)*

Participants highlighted the positive influence of elite athletes talking about their experiences of their MC, helping to accept the potential influence of the MC on performance but also for health improving help seeking behaviours for irregular menstruation:

*'I think there is a lot more chat about it now. I think everyone's starting to realise that it is actually a good thing to have your period, you have more negatives by not having it... So, I think probably all the discussion that's going on, and I know a professional mountain bike rider will post about it on Instagram, and that's probably spurred me to go and seek help [for lack of periods].' (P14 – elite)*

*'I think it's amazing that world class athletes are sharing it with everyone because I remember back in the day when it wasn't very widely spoken about. I felt like there was something wrong with me when I didn't perform well in the race because of my period. I would beat myself up about it, whereas I think the more people share, the better it is for female athletes because they feel like they're not alone. I just think it's good to read about their experiences. I mean obviously it's not nice when your period negatively affects your performance, but it's good to know that you're not alone.' (P11 – sub-elite)*

*'I saw the whole thing around Emma Pallant [triathlete] because I think her period came during a race and it was actually visible that her period had come. I thought it was really good how she spoke about that afterwards. I have noticed the woman who won the Ironman World Championships in Nice this year, Laura Phillips, I heard her interview pretty much immediately after the race and she talked about how the timing of the race, she actually mentioned she knew the timing was good, that she would be feeling strong in her menstrual cycle, and I thought that was really good. It made me think, they are thinking about these things, so I am aware that it's becoming more talked about than it used to.' (P20 – club)*

**Knowledge and education on MC**

For most participants, school provided their initial education on the MC, but as recently reported, this education was very basic around menstruation, menstrual products and sex education, as opposed to management of menstruation and MC related symptoms [85]. Twelve of the participants (eight club, three sub-elite and one elite cyclists) talked about undertaking personal education either online via social media or websites, books or talking to other riders:

*'I took the responsibility to find out for myself. I wasn't on any contraception and felt that my cycles were natural, and I needed to understand why I was feeling the way that I was.' (P18 - elite)*

*'Just a little bit of reading myself. Like I said about strength training being a little bit better at the beginning of your period but no official advice.' (P15 - club)*

Six participants (four sub-elite and two elite cyclists) had received more formal education or advice on the MC regarding training and competition from either their coach, sports science practitioners, team or National Governing Body:

*'For the team, we've done some webinars on different topics, things like hormones, menstrual cycle.' (P6 – sub-elite)*

Most of the participants talked about their lack of education around the MC, especially the interaction of the MC with sports performance and management of MC-related symptoms. This agrees with findings that only a quarter of Australian cyclists rated their MC knowledge and education as good [32]. Participants highlighted that more education for athletes is needed which agrees with findings of recent studies where female athletes have expressed that they want more MC education [77,79] including 69% of Australian female cyclists [32]:

*'I have never had any education on it [MC] at all, and I've been part of a go ride club. I've been part of teams. I've really been in the system.' (P4 – sub-elite)*

*'I think even a lot of women don't understand it [the MC] … especially younger women and girls coming into sports I think they need to know from an early age of about what's healthy and what's not.' (P3 – club)*

The poor quality and lack of MC education starts in schools [85]. Most females are educating themselves, although caution needs to be exercised as social media, websites and books provide advice that is not always supported by science and best practice [86]. This is particularly important for club and recreational cyclists who do not have access to other sources of advice, as illustrated by the findings in this study that none of the club cyclists had any formal education

or advice on the MC and relied on self-education. Therefore, they need access to freely available high quality scientifically based guidance, which would enable athletes to make informed decisions and better manage their MC to optimise training and competition, and for coaches to better support female riders.

## Limitations

One of the aims of this research was to investigate female cyclist's experiences and perceptions of the MC on training and competition performance from under 23 and senior cyclists. However, we only managed to recruit one under 23 cyclist even though we contacted teams with junior and under 23 riders, therefore we were unable to conduct age-based comparisons. This may be due to younger female athletes being less comfortable discussing their MC [19]. This study has a small cohort of elite cyclists, which may not be representative of the wider cohort of elite cyclists. However, recent research has found there are similar MC experiences and perceived effects across cycling competition levels [32]. All the participants in this study were UK-based and therefore their experiences may not be representative of cyclists in other countries, from different cultural backgrounds or those who are commuter or recreational cyclists. We recruited participants through the research team's links with the cycling community and governing bodies, and therefore the participants we recruited may be more connected with the sport and governing bodies and more willing to discuss issues around the MC. Therefore, their experiences may not be representative of those who are not part of formal teams, or who are recreational cyclists, or from under-represented groups.

High quality interview data relies on the rapport between the researcher and the participant, as this influences the participants level of comfort in sharing their experiences. The lead researcher had met several participants previously but met some for the first time on the day of the interview which may have influenced the participants comfort in sharing their experiences. The participants in this study were typically older senior athletes who may be more comfortable having conversations about their MC, so this may influence the findings about openness of conversations and may not be representative of all cyclists.

The lead researcher is a female cyclist and conducted all the interviews. Her knowledge and personal experiences helped build a rapport with the participants which is important in qualitative research [87]. However, it is important to acknowledge that as a co-creator of the data, the lead researcher's personal experiences had the potential to shape the direction of the interviews and exact questioning [88]. The lead researcher was reflexive and discussed the findings and interpretations of the data with the research team to get an outside perspective.

We asked the participants in a pre-interview questionnaire to self-report their MC related symptoms and we also only interviewed the participants once about their experiences of MC and training and competition. Therefore, due to the retrospective nature there is the possibility of recall bias [89]. Future studies could combine participant interviews and menstrual cycle symptom tracking for several menstrual cycles as utilised by [48] to get a more complete understanding, and minimize recall bias.

## Practical implications and recommendations

- Improve education for cyclists and coaches, via accessible educational materials (e.g., national cycling governing body websites) on the MC and its potential effect on training and performance, MC related symptoms and symptom management (particularly non-pharmacological), menstrual products, HC options and menstrual disturbances including the association with REDs for athletes, coaches and medical professionals.

- Cyclists may benefit from symptom tracking across multiple MC cycles to better understand any negative interactions with training and performance to enable development of an individualised plan and implement strategies to manage symptoms experienced where necessary.

- Focus on openness of conversations between cyclists, coaches and support team. Provide guidance for coaches and support staff on how to broach discussions about the MC with athletes, equip them with the correct terminology in

coaching qualifications, and provide tangible recommendations of how to integrate MC considerations into coaching practice. For example, including information on the MC for any new riders, dedicated information sessions on the MC for riders to promote conversations, and recommendation that coaches make sure menstrual products are available for riders.

• Coaching knowledge and awareness should be improved on specific issues related to the MC for female cyclists when planning training programmes, e.g., potential for increased saddle soreness during menses and the challenge for riders locating toilet facilities on long outdoor rides during menses to change menstrual products.

• Participants requested more research in sports science including the MC symptom management, female physiology and training for female cyclists to allow them to optimise their training and performance.

## Conclusion

Female cyclists' experiences and perceptions of the MC on training and competition performance were wide ranging with participants reporting a range of MC symptoms that varied in severity and timing in the MC. 45% of the cyclists had experienced irregular menstruation either currently or in the past with participants suggesting possible causes may be due to low body weight and hard training with insufficient energy intake. 73% of the participants spoke to their coach about their MC but typically this was limited. No participants undertook MC-related symptom tracking across the MC even though for some participants these symptoms were perceived to affect training and competition. Individualised training informed by athlete symptoms and perceived effects of the MC is recommended. This is highly likely to be linked to the fact most participants said they lacked knowledge and education about the MC, the management of MC related symptoms to optimse training and performance. The findings highlight the need for improved education on the MC for cyclists and coaches to improve performance and athlete health and wellbeing and improve openness of conversations.

## Supporting information

**S1 Text. S1 File. Pre-interview questionnaire. S2 File.** Interview guide. **S3 File.** Interview transcripts.
(ZIP)

## Author contributions

**Conceptualization:** Louise Burnie, Paul Ansdell, Elisa Pastorio, Kirsty M Hicks, Neil Heron, Natalie Brown.

**Data curation:** Louise Burnie, Paul Ansdell, Georgia Allen-Baker, Natalie Brown.

**Formal analysis:** Louise Burnie, Paul Ansdell, Georgia Allen-Baker, Kirsty M Hicks, Neil Heron, Natalie Brown.

**Investigation:** Louise Burnie.

**Methodology:** Louise Burnie, Natalie Brown.

**Project administration:** Louise Burnie, Natalie Brown.

**Writing – original draft:** Louise Burnie, Natalie Brown.

**Writing – review & editing:** Paul Ansdell, Georgia Allen-Baker, Elisa Pastorio, Kirsty M Hicks, Neil Heron.

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
