## [Decision Letter · Decision Letter 0]

9 Apr 2026

PONE-D-26-07217Female cyclists perceived effects and experiences of the menstrual cycle on training and performancePLOS One

Dear Dr. Burnie,

Thank you for submitting your manuscript to PLOS ONE. After careful consideration, we feel that it has merit but does not fully meet PLOS ONE’s publication criteria as it currently stands. Therefore, we invite you to submit a revised version of the manuscript that addresses the points raised during the review process.

We look forward to receiving your revised manuscript.

Kind regards,

Ratko Peric, PhD

Academic Editor

PLOS One

Journal Requirements:

2. In the online submission form, you indicated that “Data can be made avaiable on reasonble request.”.

Reviewers' comments:

Reviewer's Responses to Questions

**Comments to the Author**

1. Is the manuscript technically sound, and do the data support the conclusions?

Reviewer #1: Yes

Reviewer #2: Partly

2. Has the statistical analysis been performed appropriately and rigorously? 

Reviewer #1: N/A

Reviewer #2: N/A

3. Have the authors made all data underlying the findings in their manuscript fully available?

Reviewer #1: Yes

Reviewer #2: No

4. Is the manuscript presented in an intelligible fashion and written in standard English?

Reviewer #1: Yes

Reviewer #2: Yes

5. Review Comments to the Author

++++++++++++++++++++++++++++++++++++++++++

Reviewer #1: Overall

This manuscript draft explores female cyclists' experiences with menstrual cycle (MC) effects on training and performance through qualitative interviews. It identifies key themes like symptom variability, MC disturbances linked to energy deficiency, limited coach discussions, and education gaps. This manuscript was a pleasure to read, therefore congratulations to the authors on the quality of writing. This future published article will have a strong contribution to women's exercise physiology—fills the specific cyclist MC gap and supports health/performance optimization.

The following are queries and perhaps suggestions that may increase readability. They are suggestions to limitations that could be added to make them more explicit.

1. Scope of qualitative analysis

The use of a single primary coder with critical friend checking is appropriate, but the authors might consider adding one clause acknowledging that having one primary analyst may limit interpretive breadth, even though team discussions and an audit trail enhanced the study’s trustworthiness.

Line 121

The inclusion of a positionality paragraph is noted and appreciated. The authors might clarify how the lead researcher’s identity as a woman specifically informed her perspective when conducting MC research. While Line 131 acknowledges that she is female, the manuscript does not elaborate on how this standpoint may have shaped data interpretation or participant interaction

Clarification related to Sparkes et al. (2013)

For the reader there may be some confusion, possibly due to unfamiliarity with Sparkes et al. (2013)as it relates to Braun et al. (2006).

Line 189

It is unclear whether the authors are following the exact version of thematic analysis “outlined by Sparkes et al. (2013) based on Braun et al. (2006).” The described Phase 3—“searching for themes”—is worded as if themes pre-exist and are simply “found” in the data. This framing contrasts with Braun and Clarke’s constructivist approach, where themes are actively generated by the researcher. If this phrasing is directly drawn from Sparkes et al. (2013), the authors could clarify that distinction.

Line 207

The authors state that they aimed to “minimize personal bias.” However, in Braun and Clarke’s approach (and many other qualitative paradigms), bias is not viewed as something to eliminate. Rather, researcher subjectivity and reflexivity are acknowledged as integral to the analytic process. The authors might consider rephrasing this to better reflect that perspective.

Line 774 and similarly Line 764

If the phrase “attempt to minimize bias” is indeed drawn from Sparkes et al., the authors could clarify this and, if appropriate, note how their analytic approach aligns or diverges from Sparkes et al. (2013). Similarly, Line 764 suggests that personal experiences shaping the direction of interviews should be avoided; however, in Braun and Clarke’s framework, researcher experience and reflexivity are essential components of meaning-making. A brief clarification of this distinction would help align the manuscript’s terminology with qualitative standards

2. Sampling and self selection bias

You used purposive, criterion based sampling via cycling networks and federations. Recruitment is recognized as difficult. This, however, likely over represents engaged, connected riders and those already willing to discuss MC, and under represents those with most stigma or least support. Perhaps you could add one sentence noting that experiences may differ for riders not connected to these networks or less willing to volunteer.

3. Transferability beyond competitive UK cyclists

You focus on competitive club–elite riders; however, experiences of purely recreational cyclists or other cycling cultures (e.g., low resource settings, different health systems) may differ. You already hint at this with “not representative of cyclists in other countries,” but a single phrase acknowledging limited transferability to non competitive women who cycle for transport or leisure could help.

4. Reliance on self report and retrospective recall

MC characteristics, symptoms, training adaptations, and prior HC history are all self reported and in many cases recalled over years. A short sentence could acknowledge potential recall and reporting bias and note that you did not verify symptoms or MC disturbance with clinical or hormonal measures. Perhaps future studies might usefully incorporate biological verification or prospective tracking alongside qualitative methods.

5. L 744-746 Limitations

States: One of the aims of this research was to investigate if there were differences between under 23 and senior female cyclists and their experiences and perceptions of the

relating to the MC with coaches and support staff.

The manuscript states this as one aim in the limitation but in the methods (line 115), aims are split clearly as: 1) to understand female cyclist’s experiences and perceptions of the MC on training and competition performance from an array of levels (club to elite) and ages (under 23 and senior). (2) to explore female cyclist’s comfort in having conversations relating to the MC with coaches and support staff.

Suggestion: One original aim was to explore female cyclists' experiences and perceptions of the MC on training and competition performance across levels (club-elite) and ages (under 23 vs senior), and their comfort in MC conversations with coaches/staff. With only one under-23 rider, we could not conduct age-based comparisons as intended, limiting insights into junior vs senior differences.

Line 169

Inclusion of a yes/no question for a semi-structured interview is not ideal, perhaps the authors could clarify why the question was written in that way. Same thing with question 1c in the supplementary document and a few other questions. Yes or no questions may potentially limit the way in which participants can express their experiences and therefore would be a limitation.

6. Tables

Table 1 suggestion to Title change to Participant training and racing characteristic (means, SD) by Tier level (1-3)

Table 2

The Menstrual Cycle History information in Table 2 could easily be incorporated into the written body (results line 211-212 ) and therefore does not need to be in the table. Table 3 becomes Differences across between Naturally menstruating group and Hormonal contraceptive group and Non-hormonal contraceptive (intrauterine device) group.

Table 3

Given you do not separate out by group (although it might be interesting to do so) in Table 3 suggestion to change title to: Self-reported symptoms (n, %) in across all participants. I Consider separating symptoms by naturally menstruating vs HC users (as in Table 2) to enable direct visual comparison of symptom prevalence across MC status.

7.Grammar

Line 75 Athletes often “choosing” check choose.

Line 173: it says cyclist’s, check “cyclists”

++++++++++++++++++++++++++++++++++++

Reviewer #2: This manuscript addresses an important and timely topic in sports science, particularly given the growing interest in female athlete health and performance. The focus on female cyclists and the use of a qualitative approach to explore lived experiences are valuable contributions to the literature. The manuscript is generally well structured, clearly written, and follows appropriate qualitative research methodology.

The use of semi-structured interviews and inductive thematic analysis is appropriate for the research aims, and the inclusion of participants across different performance levels adds practical relevance. The findings provide useful insights into athletes’ perceptions, particularly regarding menstrual cycle symptoms, communication with coaches, and the need for improved education.

However, several aspects could be strengthened to improve the scientific rigor and clarity of the manuscript.

First, the conclusions should be more clearly aligned with the qualitative nature of the data. Since the study is based entirely on self-reported perceptions, statements related to the impact of the menstrual cycle on training and performance should be framed more cautiously, avoiding implications of objective performance effects.

Second, the discussion would benefit from deeper integration with physiological and performance-related literature. While relevant studies are cited, the manuscript could be improved by more critically engaging with conflicting findings and providing clearer mechanistic context.

Third, the sample characteristics should be more explicitly acknowledged as a limitation. The relatively small and heterogeneous sample, along with the absence of younger athletes, may limit the transferability of the findings.

Fourth, the Data Availability Statement does not fully meet PLOS requirements. The statement that data are available upon reasonable request should be revised to ensure compliance with the journal’s data sharing policy, or appropriate justification for restrictions should be provided.

Finally, the practical implications could be further developed. While the manuscript highlights the need for education and individualized approaches, more specific recommendations for coaches and practitioners would enhance the applied value of the study.

Overall, this is a relevant and well-conducted qualitative study with clear practical importance. Addressing the points above would strengthen the manuscript and improve its contribution to the field.

6. PLOS authors have the option to publish the peer review history of their article (what does this mean?). If published, this will include your full peer review and any attached files.

Reviewer #1: **Yes:** Patricia K. Doyle-Baker

Reviewer #2: No

---

## [Author Response · Author response to Decision Letter 1]

14 May 2026

We thank the reviewers for their constructive feedback. We have tried to address all points raised, please see our answers to the more detailed feedback below. All changes to the manuscript are highlighted in yellow.

Reviewer 1

This manuscript draft explores female cyclists' experiences with menstrual cycle (MC) effects on training and performance through qualitative interviews. It identifies key themes like symptom variability, MC disturbances linked to energy deficiency, limited coach discussions, and education gaps. This manuscript was a pleasure to read, therefore congratulations to the authors on the quality of writing. This future published article will have a strong contribution to women's exercise physiology—fills the specific cyclist MC gap and supports health/performance optimization.

The following are queries and perhaps suggestions that may increase readability. They are suggestions to limitations that could be added to make them more explicit.

1. Scope of qualitative analysis

The use of a single primary coder with critical friend checking is appropriate, but the authors might consider adding one clause acknowledging that having one primary analyst may limit interpretive breadth, even though team discussions and an audit trail enhanced the study’s trustworthiness.

We have added a sentence to the data analysis section to acknowledge this (Page 10, lines 201-203).

Line 121

The inclusion of a positionality paragraph is noted and appreciated. The authors might clarify how the lead researcher’s identity as a woman specifically informed her perspective when conducting MC research. While Line 131 acknowledges that she is female, the manuscript does not elaborate on how this standpoint may have shaped data interpretation or participant interaction

We have added a sentence highlighting her lived experiences could have influenced the data interpretation (Page 6, lines 121-123).

Clarification related to Sparkes et al. (2013)

For the reader there may be some confusion, possibly due to unfamiliarity with Sparkes et al. (2013)as it relates to Braun et al. (2006).

Line 189

It is unclear whether the authors are following the exact version of thematic analysis “outlined by Sparkes et al. (2013) based on Braun et al. (2006).” The described Phase 3—“searching for themes”—is worded as if themes pre-exist and are simply “found” in the data. This framing contrasts with Braun and Clarke’s constructivist approach, where themes are actively generated by the researcher. If this phrasing is directly drawn from Sparkes et al. (2013), the authors could clarify that distinction.

Phase 3: searching for and identifying themes is how it is titled in Sparkes and Smith (2013) see page 124. However, we have added some detail about what this involves to hopefully clarify the process (Page 9, lines 182-184).

Line 207

The authors state that they aimed to “minimize personal bias.” However, in Braun and Clarke’s approach (and many other qualitative paradigms), bias is not viewed as something to eliminate. Rather, researcher subjectivity and reflexivity are acknowledged as integral to the analytic process. The authors might consider rephrasing this to better reflect that perspective.

We have changed the sentence to acknowledge their role was more to encourage reflexivity (Page 9, lines 200-201).

Line 774 and similarly Line 764

If the phrase “attempt to minimize bias” is indeed drawn from Sparkes et al., the authors could clarify this and, if appropriate, note how their analytic approach aligns or diverges from Sparkes et al. (2013). Similarly, Line 764 suggests that personal experiences shaping the direction of interviews should be avoided; however, in Braun and Clarke’s framework, researcher experience and reflexivity are essential components of meaning-making. A brief clarification of this distinction would help align the manuscript’s terminology with qualitative standards

We have reworded the sentence describing the role of the research team as critical friends and that their role was to make the lead researcher think reflexively (Page 10, lines 196-198).

2. Sampling and self selection bias

You used purposive, criterion based sampling via cycling networks and federations. Recruitment is recognized as difficult. This, however, likely over represents engaged, connected riders and those already willing to discuss MC, and under represents those with most stigma or least support. Perhaps you could add one sentence noting that experiences may differ for riders not connected to these networks or less willing to volunteer.

We have added this to the limitations (Page 37, lines 750-754).

3. Transferability beyond competitive UK cyclists

You focus on competitive club–elite riders; however, experiences of purely recreational cyclists or other cycling cultures (e.g., low resource settings, different health systems) may differ. You already hint at this with “not representative of cyclists in other countries,” but a single phrase acknowledging limited transferability to non competitive women who cycle for transport or leisure could help.

We have added a note to the end of the sentence to acknowledge this (Page 37, lines 749-750).

4. Reliance on self report and retrospective recall

MC characteristics, symptoms, training adaptations, and prior HC history are all self reported and in many cases recalled over years. A short sentence could acknowledge potential recall and reporting bias and note that you did not verify symptoms or MC disturbance with clinical or hormonal measures. Perhaps future studies might usefully incorporate biological verification or prospective tracking alongside qualitative methods.

We have added this to the limitations (Page 38, lines 768-773).

5. L 744-746 Limitations

States: One of the aims of this research was to investigate if there were differences between under 23 and senior female cyclists and their experiences and perceptions of the

relating to the MC with coaches and support staff.

The manuscript states this as one aim in the limitation but in the methods (line 115), aims are split clearly as: 1) to understand female cyclist’s experiences and perceptions of the MC on training and competition performance from an array of levels (club to elite) and ages (under 23 and senior). (2) to explore female cyclist’s comfort in having conversations relating to the MC with coaches and support staff.

Suggestion: One original aim was to explore female cyclists' experiences and perceptions of the MC on training and competition performance across levels (club-elite) and ages (under 23 vs senior), and their comfort in MC conversations with coaches/staff. With only one under-23 rider, we could not conduct age-based comparisons as intended, limiting insights into junior vs senior differences.

We have revised this section based on your suggestions (Page 36, lines 740-744).

Line 169

Inclusion of a yes/no question for a semi-structured interview is not ideal, perhaps the authors could clarify why the question was written in that way. Same thing with question 1c in the supplementary document and a few other questions. Yes or no questions may potentially limit the way in which participants can express their experiences and therefore would be a limitation.

Question 1 was used in the interviews, a) to d) were prompts for the interviewer to ask to follow up if the participant didn’t give much information from the broader initial question. These questions were not set in stone and the interviewer asked questions following up on what the participant said in the interview, and to get more detail.

6. Tables

Table 1 suggestion to Title change to Participant training and racing characteristic (means, ± SD) by Tier level (1-3)

We have added this to the table title.

Table 2

The Menstrual Cycle History information in Table 2 could easily be incorporated into the written body (results line 211-212 ) and therefore does not need to be in the table. Table 3 becomes Differences across between Naturally menstruating group and Hormonal contraceptive group and Non-hormonal contraceptive (intrauterine device) group.

We choose to present the information in a table, for table 2 as we already have lots of words and this type of information has been presented in this form in similar previous research [1, 2].

We did not ask the HC users to report symptoms, obviously on reflection this would have been useful, so can’t report differences in Table 3.

Table 3

Given you do not separate out by group (although it might be interesting to do so) in Table 3 suggestion to change title to: Self-reported symptoms (n, %) in across all participants. I Consider separating symptoms by naturally menstruating vs HC users (as in Table 2) to enable direct visual comparison of symptom prevalence across MC status.

Please see comment above.

7.Grammar

Line 75 Athletes often “choosing” check choose.

We have changed this (Page 4, line 68).

Line 173: it says cyclist’s, check “cyclists”

We have changed this to cyclists’ (Page 5-6, lines 104-107).

Reviewer 2

Reviewer #2: This manuscript addresses an important and timely topic in sports science, particularly given the growing interest in female athlete health and performance. The focus on female cyclists and the use of a qualitative approach to explore lived experiences are valuable contributions to the literature. The manuscript is generally well structured, clearly written, and follows appropriate qualitative research methodology.

The use of semi-structured interviews and inductive thematic analysis is appropriate for the research aims, and the inclusion of participants across different performance levels adds practical relevance. The findings provide useful insights into athletes’ perceptions, particularly regarding menstrual cycle symptoms, communication with coaches, and the need for improved education.

However, several aspects could be strengthened to improve the scientific rigor and clarity of the manuscript.

First, the conclusions should be more clearly aligned with the qualitative nature of the data. Since the study is based entirely on self-reported perceptions, statements related to the impact of the menstrual cycle on training and performance should be framed more cautiously, avoiding implications of objective performance effects.

Thank you for this suggestion, we have softened the language throughout the results/discussion and conclusion to avoid objective implications and note potential effects on performance, or perceptions of changes in performance (these small changes are highlighted in yellow in the manuscript).

Second, the discussion would benefit from deeper integration with physiological and performance-related literature. While relevant studies are cited, the manuscript could be improved by more critically engaging with conflicting findings and providing clearer mechanistic context.

We have added a few sentences linking to the physiology literature, and why there may be differences between group-based findings and individual athlete experiences (Page 23, Lines 439-446, and Page 24, lines 450-453).

Third, the sample characteristics should be more explicitly acknowledged as a limitation. The relatively small and heterogeneous sample, along with the absence of younger athletes, may limit the transferability of the findings.

We have added further limitations, see our response to first reviewer and yellow highlighted text in the limitations section (Page 36-38).

Fourth, the Data Availability Statement does not fully meet PLOS requirements. The statement that data are available upon reasonable request should be revised to ensure compliance with the journal’s data sharing policy, or appropriate justification for restrictions should be provided.

We have added the interview transcripts as supporting information S3.

Finally, the practical implications could be further developed. While the manuscript highlights the need for education and individualized approaches, more specific recommendations for coaches and practitioners would enhance the applied value of the study.

We have added a few practical recommendations (Page 38-39, lines 788-792). We will be conducting further research to assess how best to implement our practical implications and recommendations, without this we are not in a position to add further detail.

Overall, this is a relevant and well-conducted qualitative study with clear practical importance. Addressing the points above would strengthen the manuscript and improve its contribution to the field.

References

1. Brown N, Knight CJ, Forrest LJ. Elite female athletes’ experiences and perceptions of the menstrual cycle on training and sport performance. Scand J Med Sci Sports. 2021;31(1):52-69. doi: 10.1111/sms.13818.

2. McNulty K, Ansdell P, Goodall S, Thomas K, Elliott-Sale KJ, Howatson G, et al. The Symptoms Experienced by Naturally Menstruating Women and Oral Contraceptive Pill Users and Their Perceived Effects on Exercise Performance and Recovery Time Posttraining. Women Sport Phys Act J. 2023;32(1):wspaj.2023-0016. doi: 10.1123/wspaj.2023-0016.

---

## [Editor Report · Decision Letter 1]

18 May 2026

Female cyclists perceived effects and experiences of the menstrual cycle on training and performance

PONE-D-26-07217R1

Dear Dr. Burnie,

We’re pleased to inform you that your manuscript has been judged scientifically suitable for publication and will be formally accepted for publication once it meets all outstanding technical requirements.

Kind regards,

Ratko Peric, PhD

Academic Editor

PLOS One
---

## [Editor Report · Acceptance letter]

PONE-D-26-07217R1

PLOS One

Dear Dr. Burnie,

I'm pleased to inform you that your manuscript has been deemed suitable for publication in PLOS One. Congratulations! Your manuscript is now being handed over to our production team.

Kind regards,

on behalf of

Dr. Ratko Peric

Academic Editor

PLOS One